

# Efficiently detecting outlying behavior in video-game players

Young Bin Kim[1], Shin Jin Kang[2], Sang Hyeok Lee[3], Jang Young Jung[2], Hyeong Ryeol Kam[1], Jung Lee[3], Young Sun Kim[3], Joonsoo Lee[4] and Chang Hun Kim[3]

[1] Interdisciplinary Program in Visual Information Processing, Korea University, Seoul, Korea
[2] School of Games, Hongik University, Seoul, Korea
[3] Department of Computer and Radio Communications Engineering, Korea University, Seoul, Korea
[4] AI Lab, NCSOFT, Seongnam, Korea

## ABSTRACT

In this paper, we propose a method for automatically detecting the times during which game players exhibit specific behavior, such as when players commonly show excitement, concentration, immersion, and surprise. The proposed method detects such outlying behavior based on the game players' characteristics. These characteristics are captured non-invasively in a general game environment. In this paper, cameras were used to analyze observed data such as facial expressions and player movements. Moreover, multimodal data from the game players (i.e., data regarding adjustments to the volume and the use of the keyboard and mouse) was used to analyze high-dimensional game-player data. A support vector machine was used to efficiently detect outlying behaviors. We verified the effectiveness of the proposed method using games from several genres. The recall rate of the outlying behavior pre-identified by industry experts was approximately 70%. The proposed method can also be used for feedback analysis of various interactive content provided in PC environments.

## INTRODUCTION

It is considerably useful for game developers to have the ability to predict whether game players will react in a way that corresponds to the developers' intentions. Such predictions allow developers to verify that their intended design is correct, and this verification process can enhance the overall game (*Bauckhage et al., 2012*; *Bernhaupt, 2015*; *Isbister & Schaffer, 2008*; *Ravaja et al., 2004*; *Sánchez et al., 2012*). When games are being designed, developers benefit from checking whether game players react with excitement, concentration, immersion, and surprise, and from whether they exhibit these emotions at the intended time. However, detecting whether game players react and exhibit these emotions at the intended time is considerably difficult (*Bernhaupt, 2015*).

To check for such reactions, a large game company (viz., one that develops AAA games) organized a quality-assurance team. The team measures game players' reactions in a sealed

Corresponding author
Chang Hun Kim,
chkim@korea.ac.kr

space called the user-experience test room (UX Room). In the UX room, the team records the test on video in a separate, isolated location to detect specific player behavior using a heart-rate monitor and electromyography (*Bernhaupt, 2015*; *Mirza-Babaei et al., 2013*). When the test participants finish playing, game developers and analysts gather to observe the feedback from the participants, after which a survey is conducted (*Bernhaupt, 2015*; *El-Nasr, Drachen & Canossa, 2013*; *Vermeeren et al., 2010*).

When analyzing such data, the core content concerns the degree of interest and the players' concentration. Detecting outliers based on a time series, such as joyful or bored reactions at unexpected times, provides crucial feedback for game development. Based on this feedback, games can be redesigned during their development and even after their release (*Bernhaupt, 2015*; *Hazlett, 2008*; *Ravaja et al., 2004*; *Tammen & Loviscach, 2010*). However, this approach requires many people over a long period, and small independent game companies, for instance, cannot afford to adopt this approach. Moreover, the process is not automated, making it difficult to understand the multiple players' behaviors.

In this paper, we define "outlying behavior" as excitement, concentration, immersion, or surprise commonly exhibited by game players at specific times during gameplay. The proposed method is not an invasive method that requires additional equipment, such as a heart-rate monitor or electromyography, to measure this outlying behavior. Rather, we propose a non-invasive method that allows this behavior to be detected automatically without additional equipment. Several factors required to detect outlying behavior in a game environment were modeled in the proposed system. Multimodal data was collected, including keyboard strokes, mouse inputs, volume adjustments, and data from game players observed with webcams.

With the proposed system, 30 facial feature points are defined based on the active-appearance model (AAM) (*Cootes, Edwards & Taylor, 2001*) using data obtained from the observation of game players and their movements. Additionally, an artificial neural network was developed based on these facial feature points (viz., the eyebrow, eye, nose, and mouth) (*Karthigayan et al., 2008*; *Thai, Nguyen & Hai, 2011*) to detect basic facial expressions such as happiness, sadness, anger, surprise, disgust, and fear (*Li, 2001*; *Picard & Picard, 1997*). In addition, keyboard strokes, mouse inputs, and auditory changes in the immediate environment were collected and used as factors to detect specific behavior. Based on this game-player multimodal data collected in the game environment and the specific behavior expected by industry experts at particular times, we designed a classifier to analyze specific behavior at particular times using a support vector machine (SVM) (*Chang & Lin, 2011*). We verified the proposal using actual games, demonstrating that specific player behavior can be classified efficiently.

The characteristics of the proposed system are as follows:

- We propose a non-invasive method that has few measurement errors and is friendly to game players. We solve an incompatibility issue related to the measuring environment using additional equipment, such that outlying behavior can be detected in a general game environment.

- We propose a new model that can detect the outlying behavior of game players during gaming. Unlike existing methods that use fragmentary analysis in a separate environment, our proposed method detects outlying behavior based on multimodal data from players during gameplay.

- With our method, it is possible to detect outlying behavior in an environment where multiple players are participating simultaneously. Compared to existing methods that analyze a considerable amount of data from only a few players, outlying behavior can be analyzed with our method even when several players participate in the game concurrently.

- We verified the significance of the proposal by applying it to actual commercial video games. Existing methods are in many cases limited to only a few particular genres, or they focus on a theoretical design for checking players' reactions. The proposed method was evaluated using popular games (e.g., Tomb Raider, BioShock Infinite, and League of Legends) from several genres.

## RELATED WORK

Research regarding the collection of player data within games and the use of this data for analysis is a field that is becoming more prevalent as the game market develops (*Bernhaupt, 2015*; *El-Nasr, Drachen & Canossa, 2013*; *Mirza-Babaei et al., 2011*; *Mirza-Babaei et al., 2013*). Player data is generally classified into survey-based opinion data and play data (or log data) from players acquired during gameplay (*Bauckhage et al., 2012*; *Bauckhage et al., 2014*; *Bernhaupt, 2015*; *Chittaro, Ranon & Ieronutti, 2006*; *Medler, John & Lane, 2011*; *Thawonmas & Iizuka, 2008*; *Wallner & Kriglstein, 2012*). To collect this data, a crawling tool is generally used, and log data can often be customized to collect a specific kind of data. Alternatively, players can be observed directly in order to analyze their behavior (*Canossa, Drachen & Sørensen, 2011*; *Drachen et al., 2014*; *Zoeller, 2010*). In this paper, outlying behavior is detected based on multimodal data—using log data to determine the players' behavior along with data collected from observation. In this section, the relevant research is discussed.

Many studies pertaining to existing games, research, and tests for determining users' reactions rely on manually collected self-reports from the players themselves (i.e., interviews, questionnaires, focus groups, etc.) or manual video analysis of experimental subjects (*Bernhaupt, 2015*; *El-Nasr, Drachen & Canossa, 2013*; *Pagulayan et al., 2002*; *Vermeeren et al., 2010*). These studies are reliable and easy to conduct. User reactions can be accurately deduced, and the analysis becomes easier as user data accumulates. Although research has been conducted by detecting outlying behavior in games based on this method, users must invest ample time in addition to game playing for these self-reports, and developers must spend a lot of time analyzing the reports. Our research focuses on an automated method that does not require additional time for analysis.

Research that involves observing users with biometric equipment to automatically detect outlying behavior has also been conducted (*Bernhaupt, 2015*; *Bersak et al., 2001*). The use of biometric data is helpful for video-game analysis (*Bernhaupt, 2015*;

*Kivikangas et al., 2011*; *Vermeeren et al., 2010*). In particular, researchers have used biometric factors such as the heart rate, skin conductance, facial electromyography, and electroencephalography (*Guger et al., 2004*; *Hazlett, 2008*; *Mirza-Babaei et al., 2013*; *Nacke & Lindley, 2010*; *Nakasone, Prendinger & Ishizuka, 2005*; *Nogueira, Torres & Rodrigues, 2013*). Some research has proposed methods for determining the user's reaction based on expanded data adopted from the field of psychophysiology (*Herbelin et al., 2004*; *Kivikangas et al., 2011*; *Lang, 1995*; *Ravaja & Kivikangas, 2008*; *Ravaja et al., 2004*; *Ravaja et al., 2006*). Such methods can predict a user's reaction objectively, continuously, precisely, and sensitively. Moreover, these predictions can be made in real-time. However, special equipment that is not used in gaming environments is required, and it is considerably difficult to process data from multiple users. This paper aims to determine the user's reaction with a non-invasive method for detecting outlying behavior.

Additionally, methods have been proposed for detecting the common outlying behavior of users by visualizing user data within a game environment based on user logs (*Bauckhage et al., 2012*; *Bauckhage et al., 2014*; *Bernhaupt, 2015*; *Chittaro, Ranon & Ieronutti, 2006*; *El-Nasr, Drachen & Canossa, 2013*; *Medler, John & Lane, 2011*; *Thawonmas & Iizuka, 2008*; *Wallner & Kriglstein, 2012*). These visualization methods aim to analyze player progress (e.g., the elapsed time and the location of a player's death) to balance the gameplay (*Bernhaupt, 2015*; *Vermeeren et al., 2010*). Such methods benefit from having access to an abundance of user data. This data is easy and straightforward to obtain. However, many such methods are limited to particular games or genres, and they cannot be easily integrated with other genres and games.

Based on this related research, we propose a method that can efficiently detect outlying behavior in a non-invasive manner, based on an automated user-observation method and log data. In 'Methods,' we explain our proposal for detecting outliers, and we describe a classifier using user data. The experiment we conducted to evaluate the proposed method is detailed in 'Experimental Material,' and the experimental results are provided in 'Experimental Results.' Finally, we discuss these results and offer our conclusion in 'Discussion and Conclusions.'

## METHODS

### Defining outlying behavior

The proposed method is designed to detect outlying behavior that is commonly exhibited at specific times by the players in a game environment. A diverse range of player data is collected in the game environment in order to detect this outlying behavior. To determine the various types of data needed, we first conducted an experiment to extract common player reactions.

We asked three AAA developers to identify the game events they considered important in two commercial games (viz., Tomb Raider and BioShock Infinite). Users showed various reactions to particular game events, and these reactions were connected to outlying behavior which was subsequently detected (*Bianchi-Berthouze, 2013*; *Bianchi-Berthouze, Kim & Patel, 2007*). To determine how players react to each of these events, we randomly

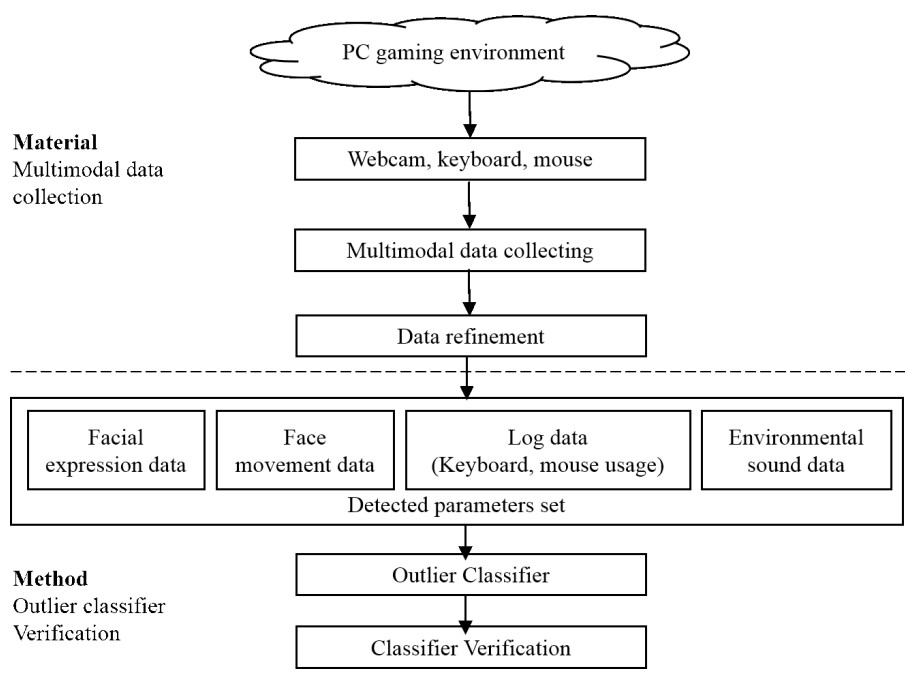

**Figure 1 System overview.** The outlier classifier is generated based on the multimodal data of users collected in a PC game environment.

selected 30 experimental subjects online. We observed the outlying behavior of these 30 subjects during gameplay. We extracted facial feature points from the corresponding experimental subjects to measure movement, keyboard and mouse usage, and volume. After playing the games, we analyzed data from normal and outlying behavior occurring at certain intervals.

Common player behavior can be summarized by noting increased body movements, facial contortions, widely opened eyes, raised eyebrows, opened mouths, blinking patterns, noises emitted by the user, and changes in the use of the keyboard or mouse. There were cases where these particular actions occurred independently, but most occurred as a composite of these actions. The complexity of these compound actions makes it difficult to check for such cases automatically. In the following section, we illustrate our method for detecting this complex behavior.

## System

To detect outlying behavior, this system detects 30 input parameters in real-time. Based on these parameters, the system utilizes a machine-learning-based outlier classifier by employing an SVM to analyze the upper level of behavior (see Fig. 1).

This system does not require the use of biometric factors such as skin conductance, facial electromyography, and electroencephalography, because our aim is to apply the proposal to a general PC gaming environment. Therefore, the parameters are detected with slightly less accuracy, but data can be collected more easily from multiple players. The types of parameters detected are listed below.

- Facial feature points: 30 feature points are extracted based on the active appearance model (AAM), and these feature points are categorized into 24 groups. Next, the angle variations of the characteristics in these 24 groups are measured.
- Facial expressions: Facial expressions are classified into six groups: neutral, happy, sad, angry, surprised, and disgusted (*Li, 2001*; *Picard & Picard, 1997*). These are classified using an artificial neural network based on the facial feature points.
- Keyboard: Keyboard usage is measured in strokes per unit of time.
- Mouse: Mouse input is measured in terms of the distance covered by the mouse cursor per unit of time.
- Volume: The relative volume is measured according to the volume of the initial environment.

Each of the above items is detailed in the following sub-sections.

### Facial feature points

To identify the facial expressions in players, this study detected facial feature points and established methods for tracking them. Research is available on the detection of such facial feature points (*Wang et al., 2014*) and it has been used in various fields (*Gu & Ji, 2004*; *Kang, Kim & Kim, 2014*; *Kaulard et al., 2012*; *Li et al., 2013*; *Pejsa & Pandzic, 2009*). There are a number of commercial solutions related to these research results (*Pantic & Bartlett, 2007*). Facial feature points are modeled using an active appearance model (AAM). AAM is one of the best-known facial feature extraction techniques, and it has been described in various works (*Cootes, Edwards & Taylor, 2001*; *Gao et al., 2010*; *Gross, Matthews & Baker, 2005*). Research involving AAM-based facial feature point detection (*Anderson et al., 2013*; *Fanelli, Dantone & Van Gool, 2013*; *Martins, Caseiro & Batista, 2013*; *Tzimiropoulos & Pantic, 2013*; *Wang et al., 2014*) has been conducted using various techniques. We detected players' facial feature points through a generic version of the AAM, rather than with person-independent AAM (*Cootes, Edwards & Taylor, 2001*). For learning, this study used variations in the 30 feature points associated with the eyes, nose, mouth, eyebrow movements, and blinking movements (see Fig. 2). The applicable modules were implemented using OpenCV (*Bradski, 2000*).

As parameters, we used the variations in movement that occur between the previous time-step and the current time-step in the 30 facial feature points during common player behavior. To analyze these variations in movement, associated feature points are grouped to measure the angle that is formed in the applicable part. The 30 feature points are condensed into 24 feature points for the calculation (all of the groups for the right eyebrow, left eyebrow, right eye, left eye, nose, and mouth are composed of an entire edge, start edge, middle edge, and end edge). The internal angle formed between three feature points connected within a set is then used. With this process, the movement of common parts that are composed of feature points and their local movements can be detected efficiently, regardless of the player's location. In the case of blinking, if the distance value between feature points at the upper and lower part of the eye is reduced to zero, this movement is regarded as a blink.

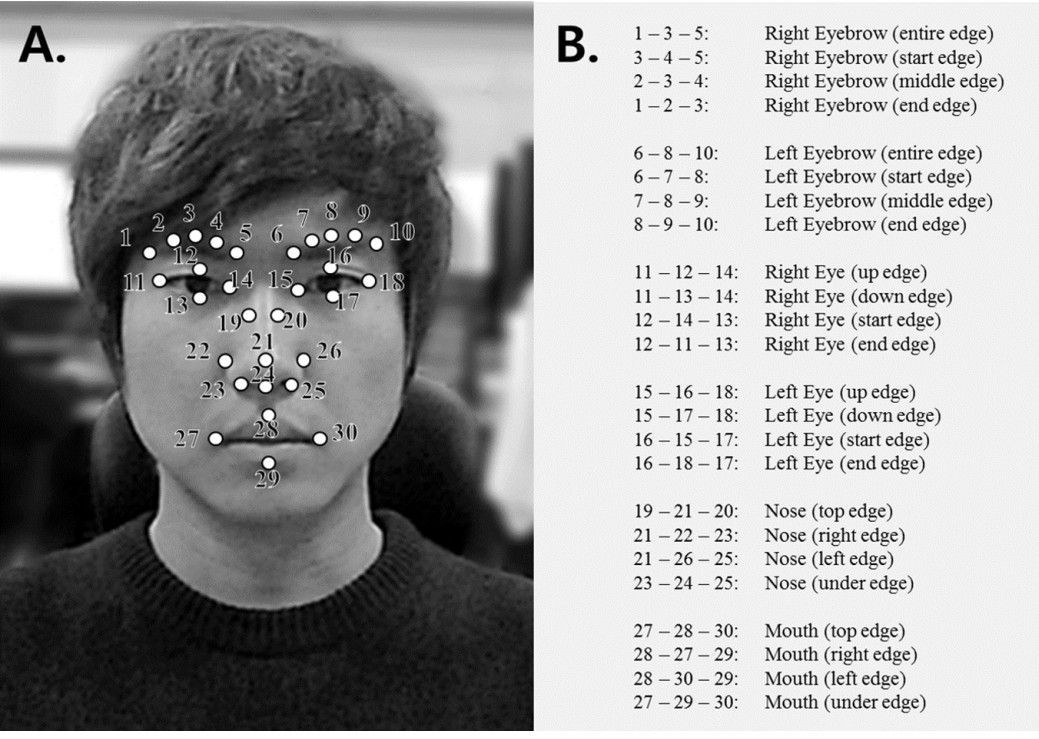

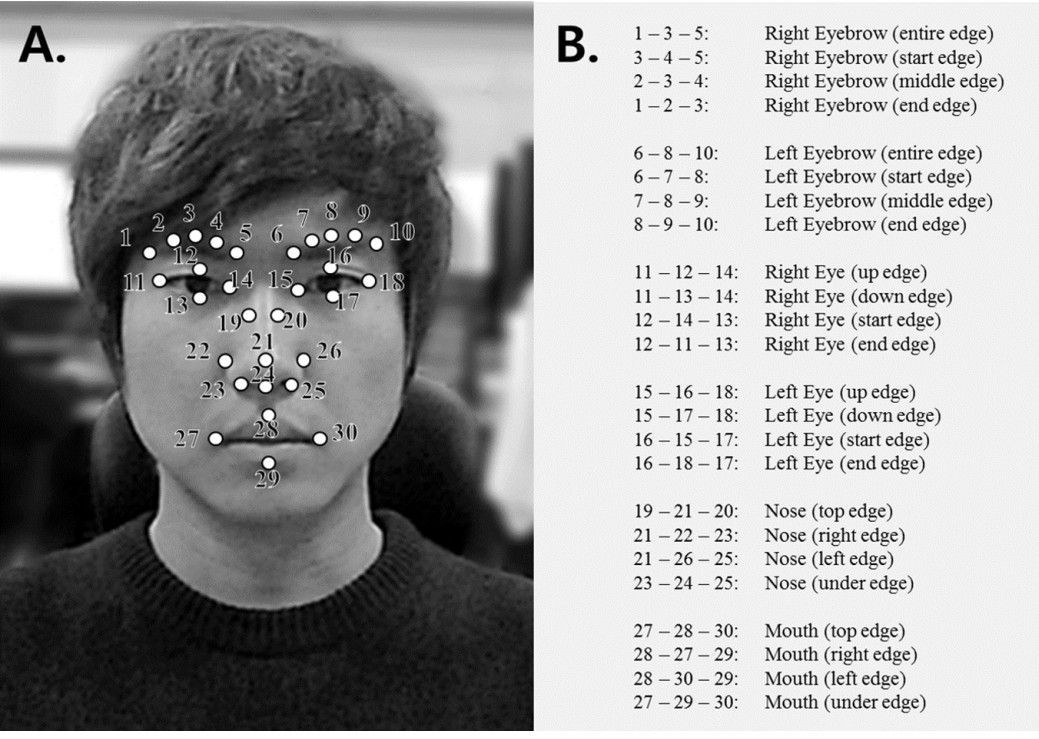

**Figure 2** **30 facial feature points and all 24 feature points constructed from these 30 feature points.** (A) 30 facial feature points used for facial-expression and blinking recognition (B) all 24 feature points constructed from these 30 feature points for measuring angle variations.

Because a player's face loses its feature points in pitch, yaw, and roll situations, parameters are not sufficiently guaranteed in such cases. AAM estimates rigid motion, but we pursued more stable data loss and aimed to prevent incorrect values from being inputted. Indeed, research has been conducted on the AAM to resolve this problem (*Anderson et al., 2013*). We used a simpler approach, because our study focuses on real-time analysis. To measure the facial rotation of players based on the feature points, we created a triangle that connects the tip of the player's nose to the sides of both eyes (Fig. 3). The aim is to identify the direction in which the player's face is rotating, by analyzing changes in internal angles of a triangle. When feature points are rotated, they are weighted to the same extent as the movement variations of the adjacent feature points for calculation.

### Facial expression

In cases involving facial expressions, we used facial feature points from previous research efforts that employed artificial neural networks (ANNs) to conduct facial analysis (*Karthigayan et al., 2008*; *Thai, Nguyen & Hai, 2011*). *Thai, Nguyen & Hai (2011)* performed edge detection using the Canny algorithm to recognize facial expressions with an ANN. *Thai, Nguyen & Hai (2011)* did not extract feature points from all faces, but instead analyzed local features that might have a direct impact on facial expressions involving the eyes, mouth, brow, and other facial areas. Similarly, we divided 30 facial feature points into five groups to recognize facial expressions. An applicable group consists

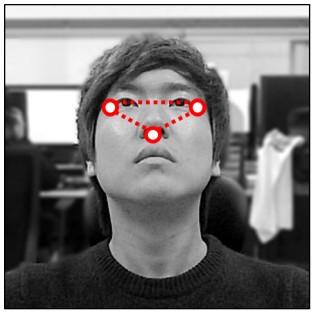
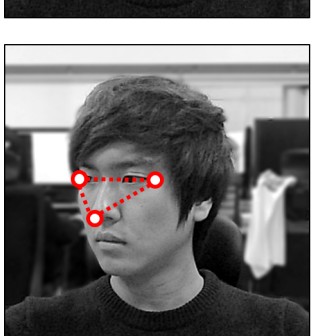
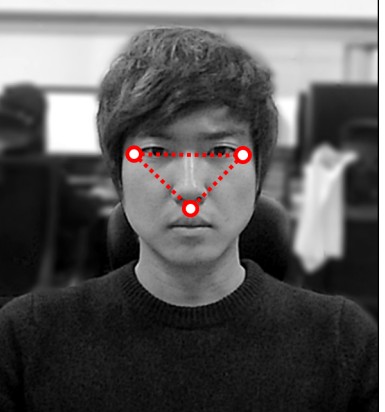
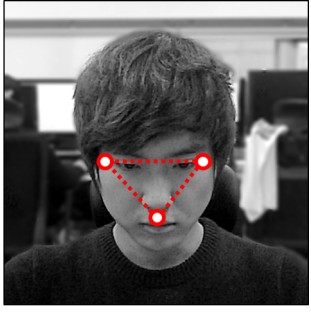
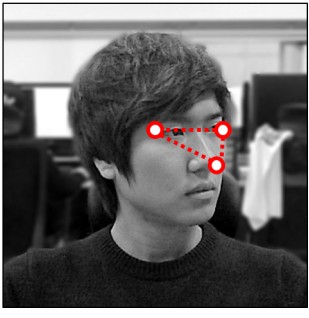

**Figure 3 To identify the direction of a player's rotation and to detect lost feature points.** A triangle is created to connect feature points on both sides of the eye and the tip of the nose. The direction of rotation is defined according to the triangle's internal angles, and weights are applied if feature points are lost in the applicable state.

of feature points that may have the largest impact on changes in facial expressions in general (*Kanade, Cohn & Tian, 2000*) (see Fig. 4). For the ANN model, we used the sum of the movement variations of the feature points that belong to the applicable group of the five facial feature points defined above in the five input neurons. The model also contained a hidden layer of 20 neurons and 7 output neurons. The output comprises mutually exclusive binary bits representing an emotion (neutral, happy, sad, angry, surprised, disgusted, and fearful). To perform ANN learning, datasets were created using training data collected from a group of ten participants. Regarding the seven facial expressions, data was collected twice from each facial expression of the participants, and this data was then used for learning. Using the datasets, tests were conducted with ten-fold cross validation to determine whether an accuracy rate of 70% or greater could be achieved. Based on these results, the facial recognizer is prepared beforehand.

Our model does not require highly accurate facial feature points or facial expressions. Rather, it is designed merely to detect outlying behavior based on these features. Indeed, a state-of-the-art method could also be applied to achieve more accurate results.

### Log data and environmental sound

Log data based on the keyboard and mouse usage is significantly relevant for identifying the status of players. Using this data, keyboard and mouse patterns and usage can be recognized, and the proficiency of the user can be inferred to judge the level of difficulty. Based on the proficiency and the level of difficulty expressed by the user, the level of immersion in a particular game could be measured indirectly (*Ermi & Mäyrä, 2005*; *Kang,*

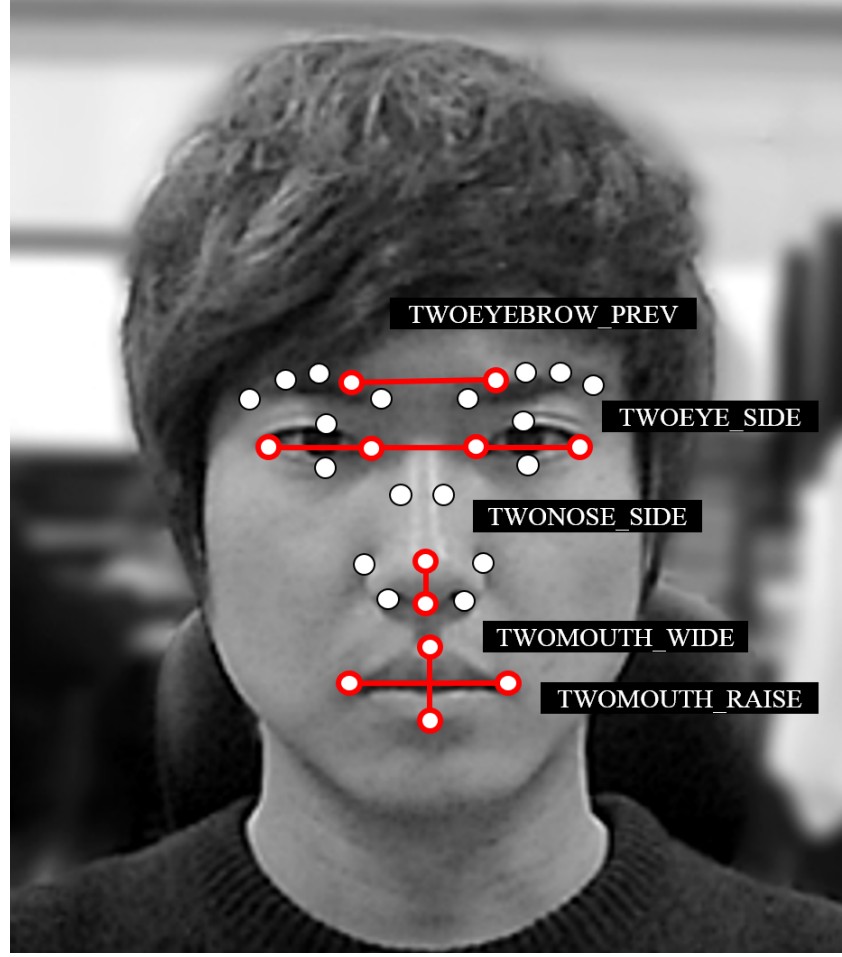

**Figure 4** **All 30 feature points divided into 5 groups.** The sum of the variations in movement for the applicable group is used as an ANN learning input node for facial-expression recognition.

*Kim & Kim, 2014*; *Kim, Kang & Kim, 2013*). This can be used to better understand how the UI is used. With our proposal, we measured keyboard and mouse usage per second during gameplay. The keyboard is used more often when players are concentrating and less often when they are frustrated. Thus, the keyboard plays an important role in detecting outlying behavior.

To record mouse usage, we measured the distance covered by the mouse cursor each second. With this information, it is possible to identify the player's concentration level with additional indirect measurements of the player's main eye points. We identified whether outlying behavior was likely to occur based on the movement of the mouse.

Finally, we measure the environmental sound. This sound can be obtained easily using the microphone on the webcam. We used the NAudio sound processing package to measure the volume with the microphone, and we recorded the change in volume from its initial value. The impact of sounds was relatively small, but it was nevertheless considered because it represents a significant component of the users' reactions.

*Classification*

Learning was performed with an SVM based on player parameters that belong to marked events, along with player parameters extracted in other frames for these parameters. Our aim is not to analyze the intricacies of the players' behavior, but rather to establish the times during gameplay when certain outlying behavior is exhibited. Therefore, it is appropriate to use a SVM. The data we used for learning consisted of a few instances of outlier data, and a larger volume of general behavior data. To balance the data, we used the Synthetic Minority Over-sampling Technique (SMOTE) (*Chawla et al., 2002*) to perform oversampling on the outlier data during the learning process. We used LibSVM (*Chang & Lin, 2011*) and RBF Kernel ($K(x, y) = \mathrm{e}^{-\gamma \|x - y\|^2}$)-based learning. Using LibSVM, the learning data was cross-validated in order to search for optimum parameters for the RBF Kernel. This method is exceptionally efficient, based on a relatively small amount of learning data. Although this study uses an SVM, we believe that the use of other two-class learning techniques or classification algorithms can produce similar results. The outlier classification process is designed to be customizable.

## EXPERIMENTAL MATERIAL

To conduct the experiment, a total of 18 participants were recruited. All participants were adults of 19 years or older, and acquired the games for the experiment from the Steam library (http://store.steampowered.com/). However, none of the participants had played these games before, and players with a Level-30 League of Legends account (http://na.leagueoflegends.com/) were selected. The participants were sought online with anonymous participation requests, and they were allowed to inspect the program for online system verification. The program was automatically implemented to collect multimodal data from the participants (keyboard and mouse usage, game screen, facial expression, volume, and player movement) during the game used in the experiment. The data was collected after the participants finished playing the game. The timeline of actual games was detected using the game screen, and only data corresponding to particular events was used in the experiments. The 18 participants were instructed to provide learning data for the individualized outlier classifiers in advance of the experiment. Tomb Raider—an action-adventure first-person shooter (FPS) genre—was played for 25–30 min in order for the outlier classifier to learn individually. This reduced deviations between individuals that occurred mainly during the emotive tests administered in advance.

To acquire learning data, we utilized data obtained during previous gameplay. And facial signals and player log data that appear during those times were used as learning data. These events where outliers are likely to occur were selected in advance by experts in the field (viz., AAA developers). Identical gameplay videos were played five times by three developers, and the expected user reactions were checked during the events. Events that were marked twice or more were judged as events where outliers were likely to occur. However, for the multiplayer online battle arena (MOBA) genre, where the same game results cannot be guaranteed among all players, the time at which the player dies was checked. Further data regarding the correspondence of the timelines can be found in Data S1.

To constitute a learning group, approximately 2.5 s of data was marked as an outlier event, beginning at the point where the outlier occurs. Other non-outlier events were marked as general behavior. During non-outlier events, abnormal behavior can occur, and such behavior can reduce the learning accuracy. However, the impact of such behavior is relatively minor, judging from our experiments. Based on the applicable video, facial signals were detected within a 1 s time window, using the methods explained above. Prior to detection, the player's initial states were stored in 5 s increments. Empirically, the time unit was set to 5 s because there were an unacceptable number of errors in the data itself when the initial state was set to less than 5 s. The relative angle variations per second among the 24 facial feature points were used for learning, beginning from the initial state. Additionally, facial expressions, the frequency of blinking, player log data, and the sound volume were used for learning.

The 18 participants were divided into three groups prior to the experiment. To collect the data required for prediction, the groups were allotted 15 min, 25 min, and 45 min, respectively. Each group played Bio Shock Infinite from the action-adventure FPS genre, and League of Legends from the MOBA genre. By assigning the participants to play games from different genres, the experiments resulted in responses that were more diverse.

## EXPERIMENTAL RESULTS

The system was created using the methods explained above. Then, four different experiments were conducted. The first experiment was devised to verify the performance of the individualized classifiers. The second and third experiments were designed to determine the efficiency of the outlier detection using other players' classifiers, rather than individualized classifier customized for the player. The fourth experiment was designed to determine whether the individualized outlier classifier provides detection efficiently with different game genres. The respective aims of the experiments are summarized as follows (see Fig. 5):

Experiment #1: Verification of the individualized outlier classifier.
Experiment #2: Verification of outlier classification using classifiers from other players.
Experiment #3: Verification of an outlier classifier integrated for all players.
Experiment #4: Verification of outlier classification in games from other genres.

The first experiment was intended to determine whether the outlier detection was conducted properly, using the individualized outlier classifiers trained in advance by the participants. Using these classifiers, an experiment was conducted to find outliers among players who played a game from same genre (viz., BioShock Infinite). In the game timeline that was provided by experts (as explained above), the expected outlier events were pre-defined to determine whether these events matched the events that the proposed system predicted. If matches occurred one or more times within the pre-defined 2.5 s outlier event, these were regarded as successfully predicted. When outlier events were predicted based on individualized outlier classifiers, there were slight differences between each group. The group that collected learning data for 15 min achieved a recall rate of 73.31% (SD = 10.72). The group that collected learning data for 25 min achieved a recall

## Experiment #1
### Verification of the individualized outlier classifier

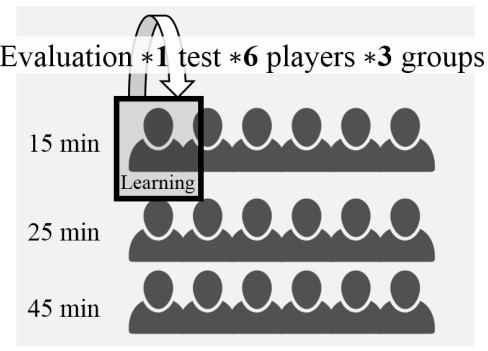

## Experiment #2
### Verification of outlier classification using classifiers from other players

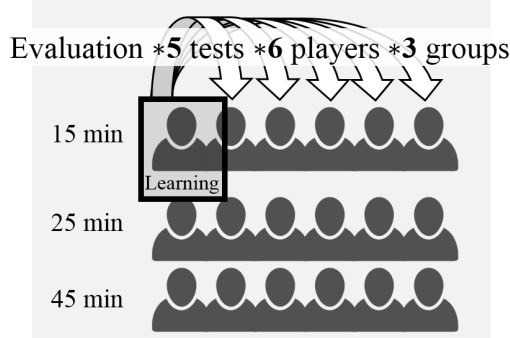

## Experiment #3
### Verification of an outlier classifier integrated for all players

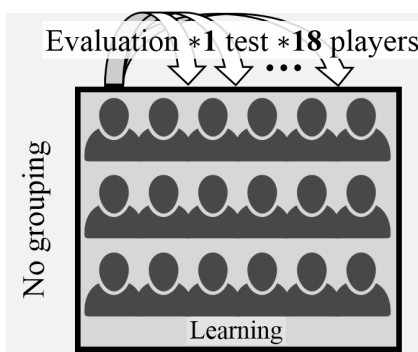

## Experiment #4
### Verification of outlier classification in games from other genres

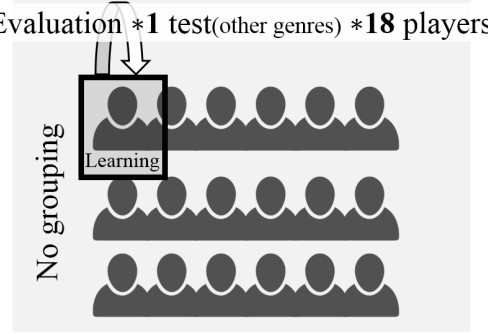

**Figure 5  Experimental design overview.**

rate of 75.29% (SD = 5.4). The group that collected learning data for 45 min achieved a recall rate of 77% (SD = 4.64). These results are listed in Table 1. Thus, better results were obtained as the learning period increased. Participants who generated low recall rates tended to be "poker face"-type players with unusual reactions during gameplay. Such players constituted an extremely small fraction of the participants, and their behavior can nevertheless be detected to some extent.

The second experiment applied individualized outlier classifiers to other players. The purpose of this experiment was to determine the efficiency of the proposed technique, even when individualized outlier classifiers were not used. In each group, one participant's outlier was detected using individualized outlier classifiers for the other five individuals within the group. Five tests were conducted for each of the 18 participants. The group that collected learning data for 15 min achieved a recall rate of 64.17% (SD = 6.73). The group that collected learning data for 25 min achieved a recall rate of 63.94% (SD = 5.43). The

**Table 1** Results for experiments 1–4. Summarization data table for Experiment #1 through #4 Results.

| Experiment | Manipulation | | | Result: recall value | | | |
| | Game genre | Learning | Evaluation | Grouping | | | No grouping |
| | | | | 15 min | 25 min | 45 min | |
|---|---|---|---|---|---|---|---|
| #1 | Same | Individual | Individual | 73.31% | 75.29% | 77% | – |
| #2 | Same | Individual | Other players | 64.17% | 63.94% | 68.26% | – |
| #3 | Same | Integration | Each Players | – | – | – | 66.54% |
| #4 | Different | Individual | Individual | – | – | – | 69.70% |

group that collected learning data for 45 min achieved a recall rate of 68.26% (SD = 5.98). These results are listed in Table 1. We thus concluded that the results are less accurate when individualized outlier classifiers are not used.

In the third experiment, the parameters from the classifiers of the 18 subjects were integrated into a single classifier to predict outlier events. As a result of these experiments, it was found that the outlier recall rate for the entire group of participants was 66.54% (SD = 3.69). Altogether, false-positive results increased and the accuracy of the results decreased. Moreover, outlier classifiers were created for each group corresponding to the learning process. As a result, when the outliers for individual players were analyzed, the group that was allotted 45 min of learning time produced the most accurate results. As indicated here, outlier events can be predicted to some extent, even if individual learning machines for each individual do not exist, and the accuracy of such predictions improves as the learning data accumulates.

The fourth experiment determined whether outlier detection could be performed properly for a game from a different genre (whether an FPS or MOBA game), using individualized outlier classifiers. When performing tests in these genres, log data such as mouse inputs and keyboard strokes were not used. In League of Legends, events do not occur according to a timeline. Therefore, predictions were made by marking the time of a character's death as the expected time for an outlier occurrence. When individualized detectors were used, the results revealed that, for the entire group of 18 participants, the recall rate was 69.70% (SD = 6.24). Similar results were obtained even when each participant's integrated outlier classifier was used. This suggests that the proposed technique can be utilized with various genres. Log data was not used for learning, because of mouse and keyboard patterns specific to the genre. However, this data could be used to obtain improved results if the proposal were trained in a single genre exclusively. In particular, in MOBA games such as League of Legends, the player's log data changed significantly whenever characters died. Therefore, such data utilization has the potential to produce better results.

## DISCUSSION AND CONCLUSIONS

In this paper, we proposed a player-friendly, non-invasive method for detecting the outlying behavior of players during gameplay. The proposed method detects outlying behavior in environments where several game players participate simultaneously, and

its significance was verified through experiments using actual commercial games. Our small-scale experiment demonstrated that the proposed system efficiently detects outlying behavior exhibited by individual players. The experiment showed the proposal's feasibility for reducing the manual work of game developers, although not by a considerable amount. Our system can be utilized by the gaming industry for evaluating games, adjusting the difficulty of levels, and balancing game designs. If the system is used for evaluating games, it can simplify the development process significantly.

With existing methods, subjects must be recruited to participate in an extensive experiment. Based on external observations or participant surveys, data coding and analysis must be performed by specialized analysts, and this typically requires weeks to complete. Evaluations based on our proposed system can be completed using an independent detection program that runs concurrently alongside the game in order to collect data for automatic analysis and visualization. This simplification thus reduces the time and human resources required. Furthermore, our method has fewer restrictions in terms of time and location, because additional equipment is unnecessary.

We introduced our proposal to ten professional game developers (some of whom were independent, whereas other worked for AAA companies). In general, their feedback was positive: one developer noted that the proposal "showed good effects with minimum equipment" and that he was "willing to use (the proposal) at the development stage"; another reports that "it has a lot of potential for improving game design." Interestingly, one of the developers told us that our method would be beneficial when using data selectively and for adjusting weights.

Our system can run in a web-based environment, because relatively little data is required for processing. Near-real-time processing is possible when using a classifier that is generated beforehand. Therefore, we are able to collect general player data and game-developer data. Such base data can be used to generate large-scale data that extends beyond the small-scale participant-based testing that is currently performed. Based on this large-scale data, the outlier-detection results within the game environment can be improved.

Furthermore, the parameters we used can be further optimized to improve the results. We used the same parameters for all experiments during the test. When using only facial-feature points in additional experiments, similar results were shown, compared to when all parameters were used. Moreover, the independent use of the other parameters showed a classification accuracy of 20% or lower. When using basic facial-feature points and combining other parameters, the use of all parameters showed the best result. Not all of the users showed better results when using all of the parameters, rather than a combination of $2 \sim 3$ types of parameters, but the overall results from using all of the parameters improved in terms of the overall classification accuracy of the users. When using the proposed system, the parameters can be optimized with classifiers, and parameters can be added depending on the particular situation and application in order to achieve better results.

The proposed system is also expected to expand the research results, because it can be used to analyze long-term content in games, videos, and other programs. Currently, the

system is designed exclusively for PC environments. However, the performance of smart-phones is gradually approaching that of PCs, and the former typically contain built-in cameras. Therefore, we shall pursue additional research to apply our system to smart-phones. Moreover, the use of the system can be expanded to applications for smart TVs and mobile environments. Other methods can enhance the system performance by combining other bio-signal detection methods (*Nummenmaa et al., 2014*) and by increasing the efficiency of emotive extraction. Likewise, the results can be improved using deep learning.

This technique targeted the detection of outlying behavior. However, it can also be employed in other fields, such as bio-signal-based child and youth indulgence- prevention systems. We believe that the proposed technique can be used in such fields, and that it can be expanded for applications in other fields.

Regarding future research directions, we intend to devise a method for processing data that is continuously accumulating, and to design system improvements based on this data. To utilize such accumulating data, we shall improve the current algorithm-reliant supervised learning system such that it can use a semi-supervised learning algorithm. By exploiting existing log-data visualization studies, we shall investigate a means for predicting the status of players (i.e., character deaths, elapsed playing time, etc.) with the ability to predict outlier events and without the need for experts to index the game's timeline. Further, although the player's keyboard and mouse input data focuses exclusively on usage, improvements shall be designed to identify the player's game situation by analyzing player chats using repetitive pattern analysis.

## ACKNOWLEDGEMENTS

The authors would like to thank the editor and anonymous reviewers for their valuable comments and suggestions to improve the quality of the paper.

### Funding

This research was supported by a Korea University Grant and Basic Science Research Program through the National Research Foundation of Korea (NRF) funded by the Ministry of Education, Science, ICT and future Planning (NRF-2015001196, NRF-2013R1A1A2011602, NRF-2014R1A2A2A01007143, NRF-2012R1A1A3006807).

### Grant Disclosures

The following grant information was disclosed by the authors:
Korea University Grant.
National Research Foundation of Korea (NRF): NRF-2015001196, NRF-2013R1A1A2011602, NRF-2014R1A2A2A01007143, NRF-2012R1A1A3006807.

### Competing Interests

JoonSoo Lee is an employee of AI Laboratory, NC Soft.

## Author Contributions

- Young Bin Kim conceived and designed the experiments, performed the experiments, analyzed the data, contributed reagents/materials/analysis tools, wrote the paper.
- Shin Jin Kang conceived and designed the experiments, analyzed the data, wrote the paper, reviewed drafts of the paper.
- Sang Hyeok Lee performed the experiments, analyzed the data, prepared figures and/or tables.
- Jang Young Jung performed the experiments, analyzed the data, contributed reagents/materials/analysis tools.
- Hyeong Ryeol Kam analyzed the data, reviewed drafts of the paper.
- Jung Lee wrote the paper, prepared figures and/or tables, reviewed drafts of the paper.
- Young Sun Kim conceived and designed the experiments, reviewed drafts of the paper.
- Joonsoo Lee analyzed the data.
- Chang Hun Kim conceived and designed the experiments, wrote the paper, reviewed drafts of the paper.

## Data Availability

The research in this article did not generate any raw data.

## Supplemental Information

Supplemental information for this article can be found online at http://dx.doi.org/10.7717/peerj.1502#supplemental-information.

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
