# Peer review of "Efficiently detecting outlying behavior in video-game players"

_PeerJ, doi:10.7717/peerj.1502_

## Round 0.1 · original submission · Major Revisions

The reviews are detailed and they contain a number of ways in which the manuscript needs to be revised including: a more relevant and cogent Introduction and motivation for the study; more details of the design; more details of the neural network model; and a more precise and detailed statement of the results.

Reviewer 1 ·

Basic reporting

The language must be checked by a native speaker. In fact, the manuscript is often difficult to read.
The introduction is not well structured and quite long and repetitive. There are too many paragraphs.
There should be a proper material section describing precisely the database (duration…) and setup used to record the multimodal data (keyboard, video). In the current version, this information is mixed with the results.
There should also be a proper method section describing precisely the multimodal features (keyboard, mouse, facial features, acoustics…) and the SVM model used to train and test.

Experimental design

The set of 24 points built from the 31 feature points should be defined in the text and not only in the figure. This choice should be better motivated.
An Artificial Neural Network (ANN) is used to perform expression recognition from facial features but the ANN model is not defined.
To increase the performance, I suggest the authors to use the delta and delta-delta of the facial features positions (i.e., speed and acceleration).
Support Vector Machine (SVM) is a family of supervised learning models. The kernel used for this study should be specified.
The time unit should be in second.
Sound is also used to help the classifier but the description of sound preprocessing is too sparse. A proper (sub-)section should be dedicated to each modality.
No reference to data synchronization appears in the text. In order to process and use multimodal data for classification, synchronization is a key aspect and should be reported.
The criteria chosen by the experts should be specified. It is not clear how the comparison between the experts’ judgements and the classifier output is performed.
It would be suitable to compare a video (+keyboard/mouse inputs)-based outlier recognition system with a physiological (SC, HR, EMG…)-based recognition system.

Validity of the findings

It would be suitable to state the actual number of “outlier” player in the group. In the text, it stated “an extremely small group” which is not sufficient.
It is not clear how this system would really improve developers’ jobs. The authors should demonstrate a semi-automatic outlier processing with their method really outperform manual processing. The conclusion and discussion section is a list of ideas not structured. It would be beneficial to rephrase this section to emphasize a link between ideas.

Additional comments

Building a system to detecting outlier behaviors is an interesting topic and the authors should pursue in this direction. However, the manuscript should better describe the setup, the method and the system used during the study. The authors may be interested in the following work: Deep Fusion: Combining Series of Events and Continuous Signals, Martinez & Yannakakis, ICMI 2014.

Annotated reviews are not available for download in order to protect the identity of reviewers who chose to remain anonymous.

Reviewer 2 ·

Basic reporting

The draft presents a system to semi-automatically detect instances of untypical user behavior during game play as intended by game developers. The aim is to support game design by testing whether affective events occur by players as intended. The technological approach seems sound, and the implementation of the classification experiments seems also correct, within the limits of reproducibility:
Most issues relate to the presentation of the work, not the data or methods. In particular, I found it hard to follow the motivation/introduction and relation of the work to related research. For example, in the introduction there is no specific reference on outlier detection of game users, although this is the main aim of the draft. As a consequence, a discussion of the results in light of related work is missing.
Also, the first paragraph of 2.1 is quite surprising and incoherent. It is not clear, what is actually attempted and how it should relate to the introduction.
In general, the presentation is often vague not implicit. So below is a long list of sentences/phrases I could not follow and have, IMHO, to be revised in order to actually understand/reproduce the work.

Abstract:
L. 4 “detect outlying behavior in individual players” suggests finding player who behave different than the majority of player. However, as I understand the draft, this work is about finding intended and expected affective player reactions for the majority of players, albeit on an individual level. Outliers are defined here as outstanding events in comparison to the baseline game play behavior.

1. Introduction:
I suggest splitting the introduction from related work. The current introduction lacks a stringent argument. The first paragraph holds no relevant information and should be deleted; it may hint at the motivation of the authors, but does not introduce the research idea. However, if you want to argue on the increase of the game market for the development of user data and analyzing methods/tools, please explicitly state it and refer to it (at line 89 I got this impression). Paragraph 3 should end with a clear statement of the aim of this article, thus introducing the related work presented.
As a minor issue, I do no follow your statement in line 34. As a researcher I suspect manufacturers to aim at financial success and good publicity, and if this is solely based on users’ playing the games as expected, I would have liked at reference/quote.
1. Related work:
L. 51: “a few methods are available” Either the naming of such methods with references is missing, or in this paragraph it is obscured that these methods will be presented in the following paragraphs. The two “however” (l 51, 54) irritate me understanding the point you are going to make.
L. 58: “experimenter’s observational video analysis reports” are mentioned but not further commented on. More information on this method and why you did not use it is required, as you did for self-reports in this paragraph.
L. 61,62: Self-reports are intrinsically subjective. That is the whole point of it, as users are not evaluation experts. Please revise this sentence to make clear your statement.
L. 80-84: “player data” The wording of this sentence (like others) is a little bit sloppy. In the preceding paragraphs are also about analyzing player data. You are talking about log data, so why not delete the 1st and 2nd sentence in favor of a revised 3rd? Please explain how such visualization can result in a “balanced game play”.
L. 87: Please explicitly state what “additional procedures” are. If you mean genre or game specific selection of metrics, or just the values of such metrics, please say so.
L. 89-104 This paragraph should be used for introducing the related work, not for ending it.
L. 97: “Taken together” This sentence does not summarizes the Introduction so far. Instead it introduces the aim of your empirical work. Please find a correct way to connect this paragraph to the related work. For example, it is not clear in the related work presentation that you will rely on log data or player observation. Also, the topic of finding outliers is not mentioned in the related work presented. Only in one paragraph on psychometric methods, outliers are mentioned, but it is not explicitly stated for the referenced literature, which actually includes outlier detection.

Minor:
Line 64,74,80 there are leading spaces. L. 126 (e.g.) there is a real intendation, other lack it altogether (e.g., L. 26)
L. 74: -> “There are research efforts TO ANALYSE”?
L. 83: -> “location of CHARACTER’S death”

2. Method
L. 107-108: This sentence offers a surprise: You introduced excitement only in l. 36. No results on emotional feedback are presented. The following sentence lack a logical junction: Do you mean to find outliers (2nd sentence) IN emotional feedback (1st sentence)? And why is this “necessary for analyzing player experiences”? It seems to me as a meaningful choice, but finding a lack of high affective responses or even signals of boredom may also be of use.
What are outliers according to you? L. 109: “if game players have behaved differently, in terms of timing of their reactions”, suggesting individual players who behave differently to the major group of players? Or as in l. 115-117: outstanding events in behavior common for most players? Please rephrase the whole section 2.1 starting with a consistent description of what you consider as outliers and how you measure it.
Minor:
l. 133: “to apply them” -> “them” can only refer to factors you, however, exclude. Use “to apply input parameters” instead.
L. 130-202: The approach to assess and classify facial expressions seems technically correct. However, I would have expected clear justification, why you differ from applying an existing solution. This research domain is vast, so there are data bases/benchmarks to use for training/validating/comparing your approach. In any case, this pretest has to be presented in more detail in order to evaluate it as a reader.
Please specify explicitly the use of an ANN with 3 layers (l. 191). Please argue why you aim for so called basic emotions including, in particular, disgust and fear (l. 193) and give references? How are these emotions distributed in games/genres you are covering? What do you mean with “can obtain even facial data” (l. 196)?
L. 204: This sentence needs a reference.
L. 204-213: There is already work done on using such information, e.g. by some of the authors themselves. Please refer to such work and how it helped to identify/classify user behavior in games.
L. 206: What are “order sets”? And who defines how these have been intended?
L. 215-218: The exact pre-processing and list of features have to be described for the acoustic input data, e.g. “a certain volume”.

L. 222: What was exactly “defined in a dictionary”? Which game do you refer to in “As for the game, …”?
L. 222-237: This whole paragraph is unclear and requires to have already understood Section “3. Results”. Information pieces are distributed all over the text in Section 2 and 3. Please use a new section 3 describing the training data and classification data of the four experiments (along with the real amount of data used), the exact information given by the experts (e.g. by an excerpt of the provided data for one of the games), the classifier, and the structure of the experiments; and move the real results to a new section “4: Results”. Currently, I can only examine the training data instead of relying on the text, which I definitely want to avoid. As one example, start with the experts marking time sections of interest to be analyzed: How many experts, how many/long sections, did these experts give complementary or redundant information? THEN, describe that input data from these sections are analyzed, and so on.

Experimental design

The experimental design is unclear. 18 participants “were instructed to learn individualized outlier classification in advance” by playing three games. Make clear that lines 258-265 present the acquisition of training data.
1. Experiment:
L. 267-281: “The aim was to identify the games in the same genre …” Was really the game identification the aim, not outlier detection? So, what exactly are your classes? Even Table 2 gives no information on the identity of the recognition. You used data from “Tomb Raider”, so please state this explicitly.
2. Experiment:
L. 285: “one participant’s outlier was detected” means one outlier from one participant, but I assume you mean all outliers from one of the six participants? Was the individual participant selected randomly, or was this experiment conducted on each participant with training data of the remaining five participants from each group?
Experiments 3 and 4 are more clearly presented.
In general, not only recall, but also precision should be reported.

Validity of the findings

As there are several issues with the presentation of the work, I cannot comment on the validity of the results. However, the discussion and conclusion seem to be coherent and follows my understanding of the presented work. I miss a discussion on the detected emotions from facial expression, though: Should it not be of interest to compare intended affect with perceived affect? A comparison of the results with related approach would improve the draft a lot.

---

## Round 0.2 · Minor Revisions

Thanks for your thorough revision of the manuscript. The reviewers recommend some minor revisions that need attention. Please revise the manuscript accordingly and address each of the comments in a covering letter.

Reviewer 1 ·

Basic reporting

I acknowledge the clarifications provided by the authors regarding the questions and modifications raised during the initial review of the article. The language was checked by a native speaker and the text is much easier to read.
The structure of the paper was modified. It clearly improves the quality of the paper.

Experimental design

The authors have clarified the technical missing information of the first version of the paper. However, the authors should still justify the use of 5 groups to recognize the facial expressions. It might be more interesting to use a reduction method such as Principal Component Analysis (PCA).
It would also be valuable to analyze which parameter (keyboard, mouse, facial expression…) provides more information in detecting outlying behavior. The authors might use a classifier such as the naïve Bayesian classifier or remove some features and check the classification accuracy.

Validity of the findings

The collected data is sound. The conclusions are appropriately stated. However, statistical tests should be run on the recognition rate before stating “the differences between these groups were insignificant”.

Additional comments

Building a system to detecting outlier behaviors is an interesting topic and the authors should pursue in this direction. The authors have clarified most of the flaws of the first version of the manuscript. However, there are still some typos that need to be corrected (see annotated pdf). In addition, statistical tests should be performed on the classification accuracy results.

Annotated reviews are not available for download in order to protect the identity of reviewers who chose to remain anonymous.

Reviewer 2 ·

Basic reporting

see comments

Experimental design

all issues met

Validity of the findings

all issues met

Additional comments

You present your work much more clearly and I could thus follow your arguments much better.
Especially, the exact procedure and the methods used (e.g. ANN, SVN) are sufficiently presented.

There is only one issue I would like the author to address again. The draft xx that specific classes are detected, such as “excitement”, “immersion” and so on. This is not the case, Instead you detect “outlying” behavior that is most likely caused by such user states, e.g. in line 57
“In this paper, we define “outlying behavior” as excitement, concentration, immersion, or surprise, commonly exhibited by game players at specific times during gameplay.” Or in line 157, or in line 6. As you do not use/evaluate the classes of “expected user reactions” given by the experts (Timeline_xxx.xlsx), this fact should be made clear.

Minor issues:

L 6-7: „The proposed method detects outlying behavior based on the characteristics of game players.“ Maybe I just find „outliers“ not a fitting term for your approach (you yourself also use “specific behavior”). Anyway, there is a reference missing to the preceding sentence to avoid the impression that it is about outliers from the found time of excitement etc., e.g. “… detects such outlying behavior …”?

L. 37: “this is especially difficult” which this, the latter?

L. 163ff Please specify explicitly the aim of collecting data from these 30 gamers.

L. 287-280 Please describe briefly how to separate immersion from difficulties from log-data for the reader without consulting the references.

L.315-316 Do you have more information on the participants to report, e.g. age, gender and gaming experience?

Figure 5: the font size of the text at the arrows will be too small

---

## Round 0.3 · accepted · Accept

Thanks for the further revision and response to the reviews.